# Adapting an Icelandic morphological database to Faroese

**Kristján Rúnarsson**
Árni Magnússon Institute
for Icelandic Studies
Reykjavík, Iceland
krunars@gmail.com

**Kristín Bjarnadóttir**
Árni Magnússon Institute
for Icelandic Studies
Reykjavík, Iceland
kristinb@hi.is

## Abstract

This paper describes the adaptation of the database system developed for the Database of Icelandic Morphology (DIM) to the Faroese language and the creation of the Faroese Morphological Database using that system from lexicographical data collected for a Faroese spellchecker project.

## 1 Introduction

The Faroese Morphological Database (FMD)[1] is the result of a joint project of the Árni Magnússon Institute for Icelandic Studies and the University of the Faroe Islands. The project, entitled the Insular Nordic Morphological Database Project, received funding from The Nordplus Nordic Languages Programme in 2021.

The FMD consists of entries for Faroese words (lexemes) with complete paradigms, including variants. Various kinds of metadata are included. It is based on a previously existing project in Iceland, the Database of Icelandic Morphology (Bjarnadóttir et al., 2019),[2] and makes use of language data collected for a previous Faroese-language project, the spellchecker *Rættstavarin*.[3] Data from DIM is used in countless language technology projects in Iceland, including smart search engines, spellchecking and hyphenation tools, taggers and parsers, speech recognition tools, online word games, and DIM is also a popular online resource for the general public. It is hoped that the new Faroese sister project will grow to be as successful in spurring the development of language technology in the Faroe Islands and aiding the general public, researchers and language students in the use and study of the Faroese language.

### 1.1 Goals

The aim was to publish the FMD with the available lexical data from *Rættstavarin* as well as the list of given names published by the Faroese Language Council.[4] The basic features of the DIM system were used to generate all inflected forms, displaying searchable inflectional paradigms on the web and providing data for download, including all the inflected forms with POS tags, lemmas and basic metadata.

Secondary goals included adding more metadata such as tags for specific morphological, syntactic and pronunciation features, dialects, etc. Recent additions to the DIM system were also tested, in anticipation of their future use for Faroese.[5]

Ultimately, the FMD should include all extant forms of all words in the Faroese language, and they should include as much useful metadata as possible. Of course "all words" is a utopian ideal as languages are constantly evolving and more vocabulary is both created and discovered, but it is feasible in the relatively near future to have basically added all vocabulary from available digital texts and to have a pipeline for semi-automatically adding newly discovered vocabulary on a regular basis. In this initial project period we focused on readily available data from lexicographical sources.

---

[1] https://bendingar.fo

[2] https://bin.arnastofnun.is/DMII/

[3] *Rættstavarin* is available as part of the Divvun language tool package at https://divvun.org/, and the source code is available on GitHub: https://github.com/giellalt/lang-fao; a description of the project (in Faroese) may be found here: https://www.setur.fo/fo/setrid/almennar-taenastur-og-grunnar/raettstavarin/

[4] http://malrad.fo/page.php?Id=38&l=fo

[5] See the description of the classification system in Bjarnadóttir et al. (2019).

## 2 Linguistic similarity

Faroese and Icelandic are closely related, both being North Germanic languages of the West Scandinavian branch and share many features such as three grammatical genders, masculine, feminine and neuter, and the four-case system of nominative, accusative, dative and genitive. Although the genitive is used much less in Faroese than Icelandic, it certainly exists and is morphologically similar. Nouns have inherent gender, while adjectives and determiners inflect for gender. Verbs inflect for mood, tense, person and number (Thráinsson et al., 2012). A full list of inflectional categories will be provided on the FMD website, in the same manner as on the DIM website.

Due to these similarities it was evident from the start that all the tools and methods that have been developed for DIM could be applied to Faroese with only minimal changes; even the web interface can be presented in much the same way, with Faroese linguistic terms simply replacing the Icelandic terms for e.g. *singular*, *nominative*, *comparative*, etc. At this initial stage of the project, the focus was on the main features of the system, though detailed tagging was employed for some particularly important or interesting morphological and pronunciation features.

The database system for the FMD is run on a copy of the DIM system. More or less the complete software system from DIM has been set up for the FMD. The system includes the database backend, import tools, and website, with both online lookup and export functions for language technology projects. A detailed description of the system may be found on the DIM website.[6]

## 3 Building the database

The premise of the project was to make use of existing data, and by far the largest set of lexicographical data available was the data from *Rættstavarin*. It, in turn, is largely derived from data from the electronic version of the Faroese dictionary (Poulsen, 1998; web version 2007, currently available at `sprotin.fo`). Another piece of low-hanging fruit was the official Faroese Language Council list of given names.

### 3.1 System comparison

The spellchecker data has words categorised by inflectional category according to a classification scheme which was created for the electronic version of the Faroese dictionary and slightly modified and expanded for the spellchecker. The spellchecker software has a template-based system that generates inflected forms from source files containing a lemma, a single template parameter and the name of the appropriate inflection pattern using a template for each pattern.

The FMD (and DIM), somewhat similarly, uses a template-based system to generate inflected forms, though the conventions for parameters are different (more than one parameter may be used to represent stem variations) and a relational database system is used rather than text files. The inflected forms are then stored in a table linked to the main table containing word entries. Additionally, a set of switches enables or disables the generation of specific sections of the inflectional paradigm such as singular or plural, definite and indefinite forms for nouns, the different moods, voices and participles of a verb, etc. The first step for each inflection pattern, then, was to create a template for it. Then the list of words with that pattern from the spellchecker data could, in theory, be transformed with a simple script to the correct import format, as long as the inflectional patterns were compatible.

### 3.2 Adapted classification and error correction

Indeed, the FMD has largely followed the spellchecker's inflection classification scheme, but it has been necessary to add new patterns to account for the subtler variations in word inflections in Faroese. For example, a number of words had been assigned a pattern which correctly accounts for their most usual or regular inflected forms, but fails to account for certain variant forms, perhaps remnants of an older inflection, perhaps novel variants, sometimes dialectal forms, archaic forms or forms used in fixed expressions. Unless assigned a different inflection template, these words would therefore be missing some of their inflected forms. In other cases the templates would have produced erroneous inflected forms.

Some accidental errors were inherited from the Faroese dictionary, while some had been introduced by the spellchecker project, and many of

---

[6]See an overview of the DIM system here: `https://bin.arnastofnun.is/DMII/aboutDMII/` and information about the structure of the available data for language technology here: `https://bin.arnastofnun.is/DMII/LTdata/`

them were simply the result of choosing the wrong pattern, e.g. forgetting that a neuter noun whose stem ends in *-s* needs a pattern that doesn't add an extra *-s* in the genitive singular form, or incorrectly typing the pattern name, e.g. writing `kv6` (feminine pattern 6) instead of `k6` (masculine pattern 6). These could often be corrected by assigning the words another existing pattern, but for many words new templates were needed. In some cases a word needs a pattern of its own due to its irregularity of inflection. There were also other errors in the spellchecker data such as typos and spelling errors and incorrectly entered template parameters.

It quickly became apparent that the number of errors in the source material was too great to leave unchecked. It would also be easier to identify and correct them early on while still working with the data in text files, rather than risking overwriting subsequent edits to database entries, particularly comment fields and other metadata, by updating them en masse later on.

The database system also requires that words be designated as base words or compounds, and a binary split point is required for compounds; e.g., the compound noun *havnarkona* is written `havnar_kona` in the lemma field to indicate that it is composed of *havnar-* and *kona*. Compounding had been indicated to some extent in the spellchecker data, but haphazardly and also with some errors.

These factors led to the conclusion that all words needed to be reviewed manually, though often somewhat cursorily due to time limitations, chiefly focusing on splitting compounds and checking for obvious errors. Along the way, tagging of morphological, usage and pronunciation characteristics was begun, and it was considered desirable that certain of them should always be tagged if possible, in particular: restriction of a word to a region or dialect; archaic, obsolete or rare usage; irregular correspondence of spelling and pronunciation; and unusual word formation patterns. This became a secondary goal of word review and, while it made it somewhat more time-consuming, it reduces the need to run through the data a second time later on, which would be even more time-consuming, and therefore serves our long-term goals well. The delay caused by manual review meant that there was no time to gather vocabulary from more sources in this round of the project, but the data has been greatly enriched and its quality improved, so it has been well worth it.

## 3.3 Importation

Data is imported into the FMD via text files with each line containing a single word entry, and may include many required and optional database fields, including the headword, the name of the inflection template, switches to limit the paradigm, and various metadata fields. These were generated semi-automatically from the spellchecker word lists and other sources using regular-expression scripting and then manually reviewed. Templates have been created manually or sometimes semi-automatically from other templates.

### 3.3.1 Nouns

The inflection of nouns was generally fairly easy to handle as they don't have as many inflected forms as adjectives or verbs and most of their patterns were already well defined. Even so, many new patterns for nouns needed to be created. For example, weak masculine nouns had only 5 basic patterns in the spellchecker data, with 3 more mixed patterns (combinations of two basic patterns) and one pattern with an irregular variant, a total of 9. In comparison, the FMD currently has 17 different templates for weak masculine nouns. This disparity is largely due to compounds with internal inflection; e.g., *lítlibeiggi* 'little brother' (accusative *lítlabeiggja*) has a more complex inflection than *pápabeiggi* 'father's brother' (accusative *pápabeiggja*). As the FMD template system has each inflected form generated from one stem and an inflectional ending, these words usually require more "stems" than other words, to account for the changes in the first half of the compound due to its separate inflection. The Faroese dictionary had not classed these words separately from compounds with an immutable first half and the spellchecker made no provision for them, although the spellchecker project had already identified them as problematic. However, such compounds are known in Icelandic and had been dealt with successfully in DIM. The FMD has followed the DIM practice of creating a separate version of each template for internally inflected compounds where required.

### 3.3.2 Verbs and adjectives

Verbs and adjectives have many more inflected forms than nouns, both in Faroese and Icelandic, and sparse information on the inflection of these

word classes in the available sources was a problem in both projects.

Verb paradigms in the Faroese dictionary are limited, omitting first and second person singular conjugations, as well as the imperative and conjunctive (optative) moods and the present participle and the mediopassive voice. Adjective paradigms also lacked comparative and superlative forms. These were added in the spellchecker project along with expansion of verb conjugation, but the spellchecker data still contains only active voice conjugations for most verbs, and the comparative and superlative forms of irregular adjectives were not obvious.

In the FMD, the verb templates now support full personal conjugation in active and mediopassive voice and a full declension of the past participle, and full paradigms are also displayed for all adjectives. Variant forms contained in the Faroese dictionary but not found in the inflection tables or the spellchecker paradigms have been added to the FMD. Additional variant forms from textual sources, such as online media and the card index of word citations (*Seðlasavnið*)[7] at the University of the Faroe Islands, have also been added.

Two software modifications were required to support Faroese verbs and adjectives, both of which are useful for Icelandic as well. The mediopassive imperative singular (without pronominal clitic) had not previously been supported, but proved to be a necessary addition for both languages. The indefinite inflection of the comparative occurs in most Faroese adjectives and was consequently added to the system. This category also exists in Icelandic but is extremely rare.

The greater number of inflected forms of verbs, the need for expanding their paradigms and the greater number of irregular verbs than irregular nouns made the creation of verb templates more time-consuming, but on the other hand, there are over nine time as many nouns as verbs, which meant that less time was needed for review of individual verbs and that, overall, the nouns took more time.

### 3.3.3 Other parts of speech

Inflection patterns for pronouns, determiners, articles and numerals have been created based on data gathered from the relevant dictionary entries, the spellchecker data, and from the Faroese grammar

by Thráinsson et al. (2012). These word classes never had inflection tables in the dictionary, only inline mentions of inflected forms and usage examples. Their inflection is somewhat similar to adjectives, but simpler in that they lack comparative and superlative forms. In some cases their inflection is very irregular, as is also seen in the same word classes in Icelandic. These words therefore required careful review, but since there are not very many of them they were fairly easy to deal with.

Adverbs, though much simpler in inflection, only inflecting for comparison, are somewhat problematic because their comparative and superlative forms are often poorly documented. Many of them had not been included in the spellchecker data because they aren't formatted as headwords in the dictionary, being merely mentioned in entries for related adjectives and often abbreviated, e.g. the adverb *broytiliga* 'variably', mentioned as *-liga* in the entry for the adjective *broytiligur* 'variable, changeable'. Most of these have not yet made their way into the FMD either. Some adverbs are uninflected, but since adverbial (non-)inflection is not necessarily explicit in the available data, all adverbs must be carefully reviewed before adding them to the FMD database. Some of the most common adverbs have been added, but comprehensive coverage of adverbs has not been achieved yet.

Uninflected word classes are also included in the spellchecker data. These words present no problems and most of them have been added to the FMD.

## 4 Present state and future additions

Currently, the FMD contains over 73,000 entries. These include about 68,000 words added from the spellchecker word lists and about 3,000 more taken directly from the dictionary, either via dictionary data collected for the spellchecker project or manual lookup on the web, and 1,688 given names from the Faroese Language Council's name list. Several hundred words have been added from other sources such as web texts and other published texts, Wiktionary[8], and Thráinsson et al. (2012). The number of individual inflected forms in the FMD is about 2.7 million and the number of distinct word forms, i.e. unique strings or types, is

[7] https://sedlasavn.setur.fo/

[8] https://en.wiktionary.org/wiki/Category:Faroese_language

about 945,000.

The FMD currently does not cover proper names well and lacks e.g. most place names, company names and surnames. Many of these may be sourced from government lists, phone directories, etc.

Corpus data can provide further general vocabulary. The Faroese Text Collection[9] (FTC) has been used as a rough gauge of the completeness of the FMD. Although the FTC only has 1.1 million tokens, at this early stage in the development of the Faroese morphological database it yields some interesting material. The FTC contains just over 71,000 unique word forms, excluding numbers, punctuation and symbols, and currently, 59% of these are already included in the FMD, having been sourced elsewhere. The FTC can continue to provide a means of evaluating the progress of the FMD, i.e. what proportion of unique tokens in the corpus are already in the database and whether the most frequent word forms in the corpus are included, as well as provide some additional vocabulary. However, a much larger text corpus (25.1 million tokens) is now available as part of the Faroese BLARK 1.0, published in July 2022 by the Ravnur Project.[10] An even larger Faroese corpus, tagged and lemmatized, is in the planning stage, and that will presumably provide much new data as well.

We expect that there will be a number of erroneous and nonstandard forms in the corpus data. These will be handled in a similar manner to the data in DIM with a system of error analysis similar to the one described on the DIM website.[11]

## 5 Conclusion

DIM has proven to be both a useful tool for Icelandic language technology projects and a very popular resource for the general public. The hope is that the FMD will have a similar impact, both in language technology and as a general resource for Faroese. In order for that to happen, the FMD needs to continue to expand and its scope needs to be enlarged. DIM contains both descriptive and prescriptive data, with extensive grading and error analysis. These aspects are, as yet, not a part of the FMD, but hopefully the creation of a larger

Faroese corpus will lead to the expansion of the FMD to include such data.

## Acknowledgments

We wish to convey our heartfelt thanks to our collaborators both in the Faroe Islands and Iceland, without whom this project would have been impossible. Heðin Jákupsson provided the chief part of the Faroese lexical data and collaborated on preparing data for import into the FMD database. Samúel Þórisson, the database manager for DIM, has set up and managed the database system for the FMD as well as DIM and has adapted the system's capabilities as needed for Faroese. Zakaris Svabo Hansen has provided linguistic expertise for Faroese, and Trausti Dagsson designed and set up the FMD website. We also thank the Nordplus Programme Committee for having faith in the project and granting us the necessary funds. Thanks are also due to our anonymous reviewers for their helpful comments, which we have taken into account as far as possible.

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
