# OpenReview forum: "Adapting an Icelandic morphological database to Faroese"
_NoDaLiDa/2023/Conference — NoDaLiDa 2023_

### Official Review · Reviewer_YYCT · 2023-03-07
**Well designed work with clearly motivated choices**

**Rating:** 7
**Confidence:** 5

**Review:**

This paper is about the development of a Faroese Morphological Database, based on a similar database available for the linguistically related Icelandic language. The work is very well thought-through and clearly described. It was easy for me as a reader to follow the process and the results, and the choices made along the way are well motivated. In addition, it was nice to learn a little more about the characteristics of the Faroese language!

For the future, I wonder if the Faroese Parsed Historical Corpus could be of any use for widening the coverage? Especially since the authors also seem interested in archaic word forms.

Some typos and minor things:
with with complete paradigms --> with complete paradigms
needs TO a pattern that --> needs a pattern that
nine time as many --> nine times as many


**Paper Type:**

Short paper

---

### Official Review · Reviewer_B7TB · 2023-03-09
**This paper analyses the database system for the Faroese Morphological Database and data included in it.**

**Rating:** 7
**Confidence:** 3

**Review:**

The paper clearly describes the creation of the Faroese Morphological Database (FMD) adapting the Database of Icelandic Morphology and using data collected for Faroese and discusses the main difficulties of adapting one database to another. This work is important not only for the Faroese language but also for the Icelandic language.
The authors explain the linguistic similarity between Faroese and Icelandic (I propose to add explicitly that both are North Germanic languages) and describe in detail the building of the FMD. It is great that the authors have concentrated on quality improvement instead of quantity.
What is missing is an evaluation of the database itself and a few words about the achievement of the broadly described goals. And I am not sure if a separate subsection (4.1) about the Faroese Text Collection is required. It could be integrated into other sections.

**Paper Type:**

Short paper

---

### Official Review · Reviewer_BTEJ · 2023-03-10
**Adapting an Icelandic morphological database to Faroese**

**Rating:** 6
**Confidence:** 3

**Review:**

This paper describes the Faroese Morphological Database (FDM) whose structure is based in the existing DIM database for Icelandic. To be perfectly frank, this does not read like a research paper. It instead resembles an (inadequate) technical report. There are no experiments, no evaluation and remarkably few references to related work.

The authors describe several decisions which were taken during the creation of the database. For example, addition of variant forms in 3.2. These seem like useful additions but it is hard to know for sure because they are not evaluated in any way. We do not get any information on the coverage of this database on text, which I think is a sort of minimum requirement when you're dealing with lexical resources of this type.

Nevertheless, I will recommend publication because I think it is important to publish such resources for Faroese.

Notes:

L127-128 I don't know what this means: The database system for the FMD is run on a copy of the DIM system.

In Section 3.1, I think some illustrations of the database structure (potentially in an appendix) might facilitate understanding.



**Paper Type:**

Short paper

---

### Decision · Program_Chairs · 2023-03-17

Accept